# Phylogenetic and Evolutionary Comparison of Mitogenomes Reveal Adaptive Radiation of Lampriform Fishes

**DOI:** 10.3390/ijms24108756

**Published:** 2023-05-15

**Authors:** Jin-fang Wang, Hai-yan Yu, Shao-bo Ma, Qiang Lin, Da-zhi Wang, Xin Wang

**Affiliations:** 1School of Life Sciences, Xiamen University, Xiamen 361102, China; 21620200156501@stu.xmu.edu.cn; 2CAS Key Laboratory of Tropical Marine Bio-Resources and Ecology, South China Sea Institute of Oceanology, Chinese Academy of Sciences, Guangzhou 510301, China; 33120220156705@stu.xmu.edu.cn (H.-y.Y.); mashaobo17@mails.ucas.ac.cn (S.-b.M.); linqiang@scsio.ac.cn (Q.L.); 3State Key Laboratory of Marine Environmental Science/College of the Environment and Ecology, Xiamen University, Xiamen 361102, China; 4University of the Chinese Academy of Sciences, Beijing 100049, China; 5Laboratory for Marine Fisheries Science and Food Production Processes, Pilot National Laboratory for Marine Science and Technology (Qingdao), Qingdao 266237, China

**Keywords:** Lampriformes, mitochondrial genome, phylogeny, tRNA loss, endothermy

## Abstract

Lampriform fishes (Lampriformes), which primarily inhabit deep-sea environments, are large marine fishes varying from the whole-body endothermic opah to the world’s longest bony fish-giant oarfish, with species morphologies varying from long and thin to deep and compressed, making them an ideal model for studying the adaptive radiation of teleost fishes. Moreover, this group is important from a phylogenetic perspective owing to their ancient origins among teleosts. However, knowledge about the group is limited, which is, at least partially, due to the dearth of recorded molecular data. This study is the first to analyze the mitochondrial genomes of three lampriform species (*Lampris incognitus, Trachipterus ishikawae,* and *Regalecus russelii*) and infer a time-calibrated phylogeny, including 68 species among 29 orders. Our phylomitogenomic analyses support the classification of Lampriformes as monophyletic and sister to Acanthopterygii; hence, addressing the longstanding controversy regarding the phylogenetic status of Lampriformes among teleosts. Comparative mitogenomic analyses indicate that tRNA losses existed in at least five Lampriformes species, which may reveal the mitogenomic structure variation associated with adaptive radiation. However, codon usage in Lampriformes did not change significantly, and it is hypothesized that the nucleus transported the corresponding tRNA, which led to function substitutions. The positive selection analysis revealed that *atp8* and *cox3* were positively selected in opah, which might have co-evolved with the endothermic trait. This study provides important insights into the systematic taxonomy and adaptive evolution studies of Lampriformes species.

## 1. Introduction

Mitogenomes have been widely used for investigating the molecular evolution and phylogenetics of different fish species, and the mitochondrial genome structure of fish has been shown to be highly conserved [1]. Nucleotide diversity an alysis over the mitogenomes is crucial for identifying the regions with high nucleotide divergence and is particularly useful for designing species-specific markers [2,3]. Such markers are useful for distinguishing taxa of highly variable morphological characteristics. Although the mitogenomic structure is conserved, there have been several reports of gene rearrangement of the mitogenome in bony fishes [4], which have primarily been explained by tandem duplication and random loss [5]. Mitochondrial genome tRNA gene loss events have also been reported in some vertebrate species [6,7,8].

The mitochondrion is often considered the cellular “powerhouse” and the major site of aerobic respiration that produces energy within the eukaryotic cell [9]. Differences in the metabolic requirements of organisms may exert diverse selective pressures on the coding genes of the mitochondrial DNA (mtDNA) [10,11]. Previous studies have shown that the mtDNA of strongly locomotive birds and mammals underwent stronger purifying selection to maintain efficient energy metabolism than that of weakly locomotive species [12]. Additionally, endothermic species require higher energy metabolism to maintain body temperature. Thus, we hypothesized that the mitochondrial genes might exhibit strong selection along with the evolution of the whole-body endothermy trait.

Lampriform fishes have developed highly specialized features to inhabit deep-sea environments; for instance, opahs (*Lampris* spp.) exhibit “whole-body endothermy” that can enhance their physiological performance while foraging in the cold, nutrient-rich waters below the ocean thermocline [13]. Another important evolutionary transition relates to body shape, which ranges from oval-shaped lamprids and veliferids to elongated forms of ribbon-like fishes (Regalecidae, Trachipteridae) [14]. Thus, Lampriformes is an ideal group of ray-finned fishes to investigate numerous questions pertaining to morphological evolution and environmental adaptation. However, studies investigating the adaptation of Lampriformes species to deep-sea life are scarce. Lampriform systematics with six families (Veliferidae, Lampridae, Lophotidae, Radiicephalidae, Trachipteridae, and Regalecidae) [15,16] has been limited by a lack of biometric and molecular data. Lampriforms are relatively rare in nature, with few specimens stored in museums [17]. Their classification underwent several rearrangements in the past. For example, the rare deep-sea family Stylephoridae was once included in Lampriformes, but recent molecular studies indicated that Stylephoridae fishes are more closely related to those in the order Gadiformes [18]. Thus, information on their diversity, taxonomy, and distribution is scarce, and the phylogenetics of Lampriformes has been controversial, especially their relationship with Acanthomorpha (spiny-rayed fishes including Acanthopterygii and Paracanthopterygii). Considering the disparate classifications in the literature, a phylogenetic study of the evolutionary status of Lampriformes among teleosts using systematic data and improved analytic methods is expected to be significant. The mitogenome is considered an ideal tool for performing phylogeny, classification, and exploring evolutionary footprint related to phenotypic diversification.

Here, we analyzed the mitogenomes of three lampriform fishes to (1) elucidate the phylogenetic relationship of the Lampriformes, (2) uncover mitogenomic variations associated with phenotypic differences, and (3) identify the adaptive evolution of mitochondrial genes of the whole-endothermic opah.

## 2. Results

### 2.1. Phylogeny of Lampriformes

Specimens of three Lampriform species were collected from three sites throughout the East China Sea between 2018 and 2021 via bottom trawling (Figure 1). The mitogenomes of three Lampriformes species—*Lampris incognitus* (GenBank accession No. OP979106), *Regalecus russelii* (OP979107), and *Trachipterus ishikawae* (OP979108), were first sequenced and annotated (Figure 2). The complete mitogenomes are typically circular, double-strand DNA molecules, and 17,376 bp long for *L. incognitus*, 16,538 bp for *R. russelii*, and 18,548 for *T. ishikawae*. The concatenated alignment of 13 protein coding genes (PCGs) produced a data matrix with 11,742 bp for 68 species representing 29 orders (Appendix A). Bayesian inference (BI) and maximum-likelihood (ML) analyses produced same phylogenetic topology with *Lepisosteus oculatus* as the outgroup (Figure 3; Appendix A). The average bootstrap support values for the Bayesian trees were high. This alignment contains eight previously reported lampriform mitogenomes. Acanthomorpha was divided into two lineages: Acanthopterygii and Paracanthopterygii. Acanthopterygii consisted of Trachichthyiformes, Beryciformes, and Percomorpha. Our results were consistent with earlier molecular analyses of Percomorpha clades including numerous taxonomic orders, such as Ophidiiformes, Gobiiformes, Scombriformes, and Syngnathiformes [19,20,21]. Paracanthopterygii included Zeiformes, Gadiformes, Percopsiformes, and Stylephoriformes. Both ML and BI analyses indicated that Lampriformes is a sister group to Acanthopterygii. Divergence time resulting from strict molecular clock analyses revealed that disc-shaped and ribbon-shaped Lampriformes diverged approximately 152.49 million years ago in the Jurassic period, while the Acanthopterygii and Paracanthopterygii diverged approximately 184.06 million years ago in the Jurassic period (Figure 3). Combined with data on the body morphology of lampriform fishes, we discovered that disc-shaped Lampridae-Veliferidae clustered into one clade, and Trachipteridae-Regalecidae-Lophotidae with elongated body shapes clustered into the other clade.

### 2.2. Mitogenome Organization and tRNA Loss of Lampriformes

Mitochondrial genomic organization is highly conserved among vertebrates and usually consists of 13 PCGs, 22 tRNAs, 2 rRNAs, and 1 non-coding control region arranged in a specific order. We compared the mitochondrial genome structure of three newly sequenced lampriform species (Figure 4; Appendix A).

Surprisingly, *L. incognitus* only had 21 tRNA, and further analysis revealed that it lacked a tRNA (Pro), which was not the case for *R. russelii* or *T. ishikawae*. We verified that a tRNA (Pro) was indeed lost in *L. incognitus* using a combination of Sanger and second-generation sequencing. Further comparative analyses of other Lampriformes species confirmed that tRNA loss events were relatively common (Figure 5A): *Desmodema ploystictum* and *Eumecichthys fiski* lost tRNA (Phe); *L. incognitus* and *Lampris guttatus* lost a tRNA (Pro); and *Lophotus capellei* lost tRNAs (Phe) and (Pro). The tRNA is an important part of the process of making proteins. It connects amino acids to their specific mRNA codons by acting as an adapter molecule. Based on this finding, we assessed whether the loss of tRNA would affect the usage of corresponding amino acids. We analyzed the codon usage bias of Lampriformes species and discovered that the amino acid composition and relative synonymous codon usage (RSCU) values of mitogenomes from 11 species were generally similar (Figure 5B; Appendix A).

### 2.3. Nucleotide Diversity and Positive Selection Analysis

A sliding window analysis exhibited a highly variable nucleotide diversity (Pi values) among the 13 aligned PCGs sequences of the 11 mitogenomes (Figure 6A). The *atp8*, *nd2*, and *nd6* genes showed relatively high nucleotide diversities of 0.304, 0.333, and 0.322, respectively, while the *cox1* and *cox2* genes exhibited comparatively low nucleotide diversities of 0.198 and 0.196, respectively.

We performed comparative mitogenomic analyses to uncover opah-specific positively selected genes, and two endothermic opah species (*L. incognitus* and *L. guttatus*) were selected as the foreground branch. Among the endothermic opahs, three positively selected amino acid sites in two mitochondrial genes were detected (Figure 6B). Among the three positively selected amino acid sites, two were located in the mitochondrial *atp8* (15Y, 51A) gene, and one was located in the mitochondrial *cox3* (158E) gene.

## 3. Discussion

### 3.1. Lampriformes Phylogeny

Studies based on morphological evidence experienced weak support; thus, recent studies are more based on molecular and biological evidence [22,23]. Our mitochondrial genome dataset corroborated the recently proposed phylogenetic classification of spiny-rayed fishes with major groups, such as Acanthopterygii and Paracanthopterygii showing high bootstrap scores (BS = 0.966) [24]. In our study, both ML and BI approaches identified Lampriformes as a sister group to Acanthopterygii, and the same result was obtained following the phylogenomic analysis of exon capture data [20] and nuclear gene sequences [22]. In contrast, Lampriformes formed a clade with other Paracanthopterygii lineages, which was recently reported after considering ultra-conserved elements [21,24]. In addition, previous mitogenomic studies indicate that Lampriformes is a sister to Acanthomorpha [18,25,26]. Analyzing divergence time revealed that Lampriformes diverged approximately 150 million years ago in the Jurassic period; however, the origin of Lampriformes lineages reported in the literature is controversial. For example, the age proposed for the origin of the Lampriformes varied from the Palaeogene to the Jurassic period [20,21,22,24,27]. Lampriform systematics is under continuous debate, thus our study laid the foundation for further research of these rare and important species.

Few phylogenetic studies investigating species-level relationships have been carried out on Lampriformes due to the limited available biometric and molecular data. We constructed phylogenetic trees for 11 Lampriformes species. A previous study based on mitochondrial cytochrome oxidase subunit o gene fragments reported similar conclusions [28]. Oval-shaped and deep-bodied veliferids and lamprids clustered into one clade, while elongated and ribbon-like fishes (Regalecidae, Trachipteridae) clustered into the other clade. Lampriform fishes are tremendously diverse in body shape, and the variation in form may be highly attributed to changes in elongation. Elongation is the dominant axis of body shape evolution in fish [29], and shape variation along this axis is often related to locomotion, feeding performance, habitat use, and trophic level. In conclusion, we constructed the most comprehensive phylogenetic tree of Lampriformes to date and clarified the phylogenetic relationship among all constituent families. Our findings are expected to benefit the systematic classification of fish and fish conservation efforts.

### 3.2. Mitogenomic Basis of Adaptive Radiation of Lampriformes

The absence of tRNA in teleosts has attracted little attention and has only been mentioned in a few studies (Table 1) [7,30]. However, tRNAs have been lost in five of the eleven Lampriformes species. The tRNA loss events are not common in bony fish but appear to be universal in Lampriformes. Lampriform fishes have developed highly specialized features to inhabit deep-sea environments, such as endothermy and body plan transition. Adaptive radiation is driven by ecological opportunity and increases the potential for diversification. We hypothesize that the variation in mitogenomic structure may be related to adaptive radiation. In other words, mitogenomic structure variations may be drivers of adaptive radiation. Although the codon usage bias was not shown to be significantly changed, we hypothesize that the nucleus transported the corresponding tRNA and substituted functions [6,8].

### 3.3. Adaptive Evolution of Mitochondrial Genes in Whole-Body Endothermic Opah

Previous studies have shown that the accumulation of mutations in mitochondrial genes is influenced by life history and cellular energy requirements [12]. Organisms have different metabolic requirements due to different body size, latitudinal (and therefore thermal) range and dietary habits, etc. Thus, the mitochondrion may be imposed varying selective pressures [10,11]. Of all these factors, the thermal environment is likely to exert particularly strong selection on the mtDNA sequence [31,32]. Moreover, the mitochondrial oxidative phosphorylation (OXPHOS) genes may have undergone stronger selection because they play important roles in energy metabolism [12]. A mitogenomic study showed a positive selection of the killer whale mitogenome cytochrome b (Cytb) gene correlated with temperature adaptation [33]. Given the specialized whole-body endothermy characteristics of the family Lampridae, we hypothesize that the molecular evolution characteristics of their mitochondrial OXPHOS genes may exhibit significant differences compared with other lampriform fishes. A total of three positively selected sites were detected in the mitochondrial *atp8* and *cox3* genes with two endothermic opahs (*L. incognitus*, *L. guttatus*) serving as a foreground branch. Although we cannot verify the biological effects of the substitutions in situ, all amino acid substitutions were associated with changes in local polarity that could influence overall metabolic performance, and consequently with a functional change [34,35,36]. These adaptive changes may have evolved simultaneously with the endothermic traits of opah, as the evolution of mtDNA has been linked to aerobic capacity [12].

## 4. Materials and Methods

### 4.1. Sample Acquisition, Tissue Collection, and DNA Extraction

Specimens of three lampriform species were collected from three sites throughout the East China Sea between 2018 and 2021 via bottom trawling carried out by local fishermen. Muscle samples were dissected from the dorsal part of each specimen, which were then separately stored at −80 °C. We amplified the mitochondrial *COI* gene to identify each species. A TIANamp Marine Animals DNA Kit (TIANGEN, Guangzhou, China) was used to extract DNA from muscle samples. The mitochondrial *COI* gene was amplified by polymerase chain reaction (PCR) using the following primers: FishF-COI 5′ TCG ACT AAT CAT AAA GAT ATC GGC AC 3′ and FishR-COI 5′ TAG ACT TCT GGG TGG CCA AAG AAT CA 3′.

### 4.2. Library Construction and High-Throughput Sequencing

DNA was randomly cut into fragments, and 350 bp paired-end libraries were constructed. After library construction, quantitative PCR and Agilent2100 Bioanalyzer (Agilent Technologies, Santa Clara, CA, USA) were used for quality control. The qualified DNA libraries were sequenced using the IlluminaNovaSeq6000M (Illumina, San Diego, CA, USA) high-throughput sequencing platform with 250 bp paired-end reads using the PE150 (pin-end 150) sequencing strategy.

Raw sequences (Raw Reads) obtained following Illumina sequencing were processed to obtain high-quality sequences (Clean Reads) after removing low-quality sequences and adapter contamination. All analyses described below were based on clean reads.

### 4.3. Sequence Assembly and Annotation

SPAdes v.3.15.2 was used to splice and assemble high-quality mitochondrial genome sequences [37]. The assembled mitogenome sequences were preliminarily annotated using the mitochondrial genome annotation server [38]. Blastp and Blastn were used to compare the preliminary annotation results with the coding proteins and rRNAs of the mitochondrial genomes of the reported relatives to verify the accuracy of the results and make corrections as required. ARWEN v1.2 was used to annotate tRNA. In the case of abnormal tRNA identification, the tRNA was verified again in combination with tRNAscan-SE v2.0 prediction. Finally, tRNA with unreasonable lengths and/or incomplete structures were excluded, and the secondary structure diagram of tRNA was generated [39,40]. Lastly, mtDNA circular and linear maps were drawn using Proksee (https://proksee.ca/) (accessed on 18 March 2022).

### 4.4. Phylogenetic Analyses

Phylogenetic analysis was performed based on the 13 PCGs of 67 mitogenomes from 28 teleostean orders including 11 lampriforms with a holostean (*L. oculatus*) grand outgroup. In total, 13 PCG sequences were extracted using the DAMBE software [41]. Sequences were aligned using the MAFFT software [42]. ModelFinder was used to select the best-fit model for the phylogenic analysis [43]. BI phylogenies were inferred using MrBayes 3.2.6 [44].

### 4.5. Bioinformatics Analyses

BEAST2 (BEAST v2.6.7) was used to estimate divergence times [45]. Calibration information (divergence times of *Decapterus tabl* and *Makaira mazara* are 66.9–93.5 million years, *Danio rerio* and *Etrumeus micropus* are 151.0–246.5 million years, and *L. oculatus* and *R. glesne* are 298.8–342.5 million years) was obtained from http://timetree.org (accessed on 29 June 2022). A tracer was used to analyze the .logs files and detect effective sample size (ESS) values. The ESS values all exceeded 200, indicating convergence [46]. TreeAnnotator filtered the evolutionary tree calculated by the program to improve reliability and evaluate its differentiation time.

EasyCodeML was used to analyze selection pressure; both the branch model and branch-site model were used [47]. The base composition and RSCU values were analyzed using MEGA v.11 [48]. The AT-skew and GC-skew were calculated as follows: AT-skew  = (A − T)/(A + T) and GC-skew = (G − C)/(G + C). The nucleotide diversity was evaluated using the sliding window analysis (window size = 200 bp, step size = 10 bp) in DnaSP v5.10 [49].

## 5. Conclusions

We report the mitochondrial genomes of three Lampriformes species for the first time. Through a phylogenetic analysis of 13 PCGs, Lampriformes were inferred to be monophyletic and sister to Acanthopterygii. Moreover, the Lampriformes order is a promising candidate for uncovering the mitogenomic structure variations associated with phenotypic diversification and for understanding the drivers of adaptive radiation. Positive selection analysis showed that opahs with thermostatic traits shared specific amino acid substitutions in mitochondrial protein-coding genes. Our results provide important novel insights into the systematics and adaptive radiation studies of rare lampriform species.

## Figures and Tables

**Figure 1 ijms-24-08756-f001:**
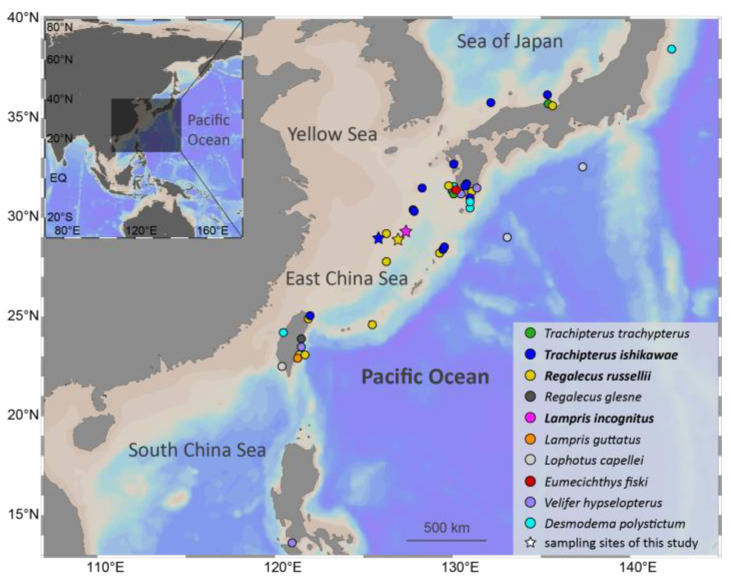
Lampriform fishes’ collection records. Stars indicate sampling locations for this study.

**Figure 2 ijms-24-08756-f002:**
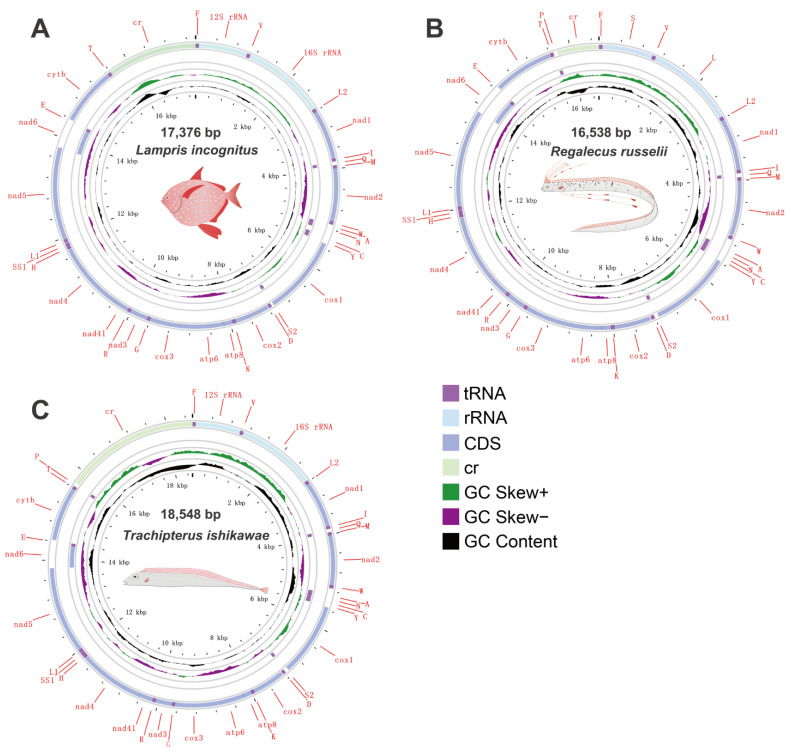
Mitogenome circular sketch map of three species sequenced in this study.

**Figure 3 ijms-24-08756-f003:**
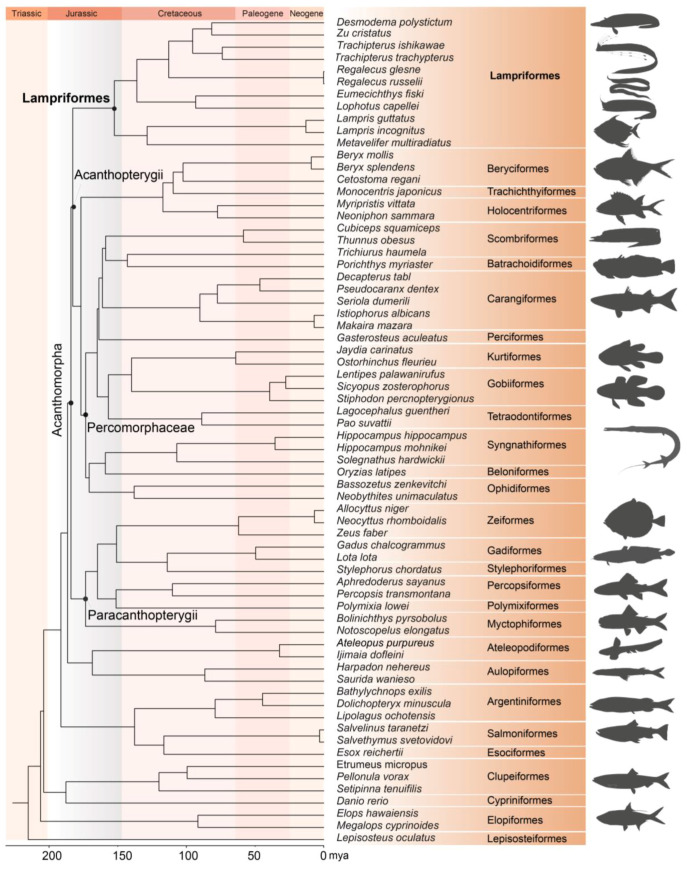
Phylogenetic tree of 68 teleost fishes constructed by Bayesian methods using 13 mitochondial PCGs. *L. oculatus* was chosen as an outgroup. Animal silhouettes are provided by PhyloPic. (http://www.phylopic.org/) (accessed on 2 November 2022).

**Figure 4 ijms-24-08756-f004:**
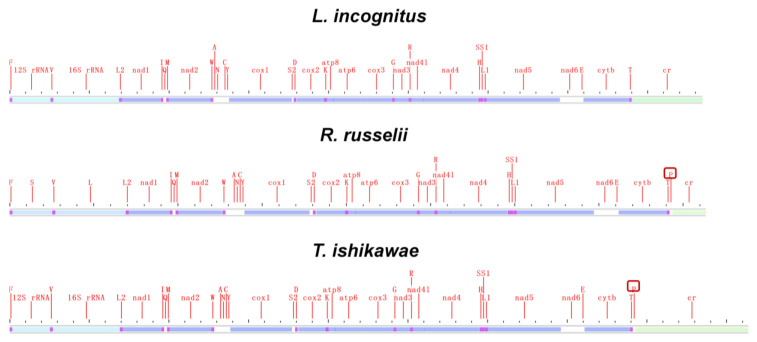
Mitogenome linear sketch map of three species sequenced in this study.

**Figure 5 ijms-24-08756-f005:**
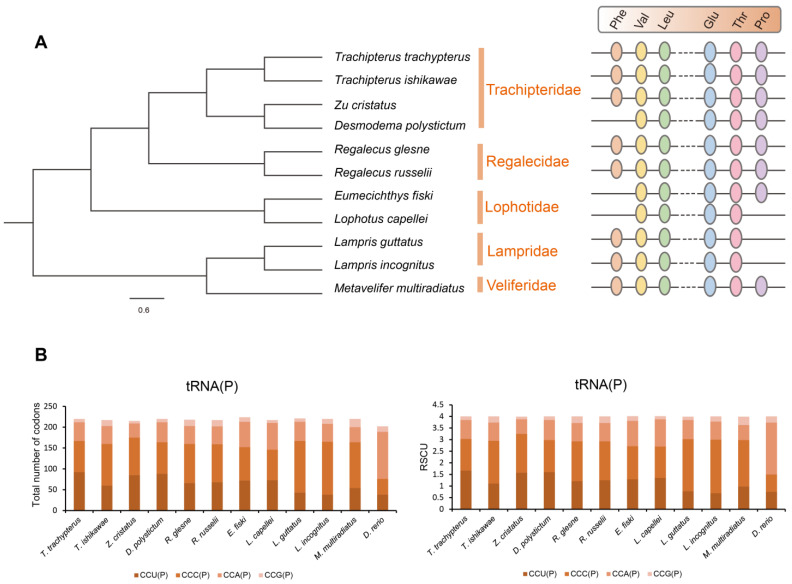
The tRNA loss in *L. incognitus.* (**A**) Phylogeny and tRNA loss of Lampriformes. (**B**) RSCU of the mitogenomes of Lampriformes.

**Figure 6 ijms-24-08756-f006:**
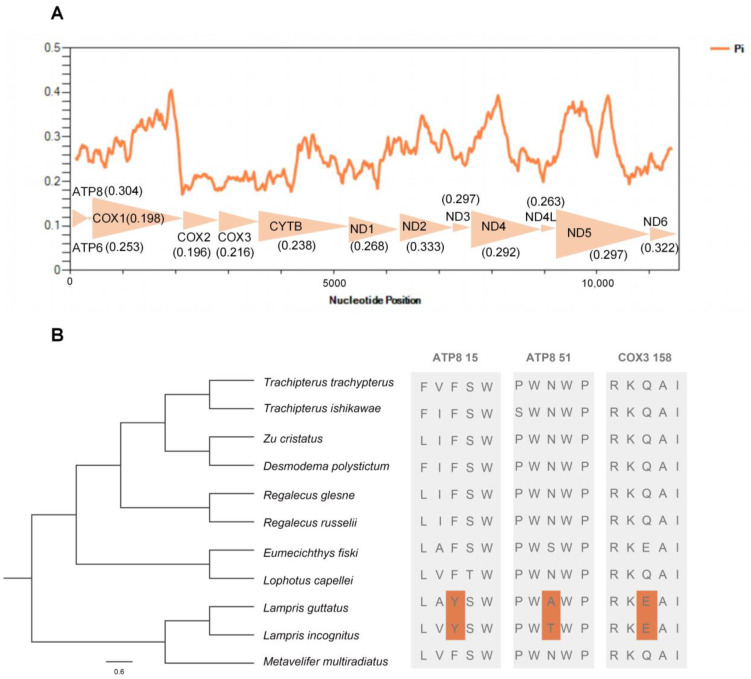
Nucleotide diversities and selection pressures of 13 PCGs in Lampriformes. (**A**) A sliding window analysis of protein-coding genes of Lampriformes. The orange curve shows the value of nucleotide diversity (Pi). Pi value of each PCG was shown in parentheses. (**B**) Analysis of the selection pressure of Lampriformes with the opah as the foreground branch.

**Table 1 ijms-24-08756-t001:** The species of tRNA loss of Osteichthyes.

**Taxon**	**Species**	**Number**	**Lost (−) tRNAs**	**Accession**
	*Chionodraco myersi*	21	−E	NC_010689
	*Lampris incognitus*	21	−P	OP979106
Osteichthyes	*Lampris guttatus*	21	−P	NC_003165
	*Desmodema ploystictum*	21	−F	AP012969
	*Eumecichthys fiski*	21	−F	AP012970
	*Lophotus capellei*	20	−(F and P)	AP012971

## Data Availability

The data presented in this study are available upon request.

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
