# Peer review of "Phylogenetic and Evolutionary Comparison of Mitogenomes Reveal Adaptive Radiation of Lampriform Fishes"

_ijms, 2023, doi:10.3390/ijms24108756_

Round 1

Reviewer 1 Report

No comment 

Reviewer 2 Report

The manuscript of Wang et al. presents the mitogenomes of three lampriform fishes used for phylogenetic tree reconstruction and comparative evolutionary study. The study is correctly designed and performed. However, some correction of the description style is required, starting from the name of the article. Since mitogenomes cannot be comparative (only the study), but compared, the title can be something like this "Phylogenetic and evolutionary comparison of mitogenomes reveal adaptive radiation of lampriform fishes".

line 55: "Thus, Lampriformes is a group of fishes that shows typical adaptive radiation." (there is no such a thing "a typical evolutionary adaptive radiation group of fishes).

line 65: the sentence should not be formulated as a question. Please, rephrase.

line 93: "Among the three species..." - the sentence is useless as the lengths of the genomes are listed in the previous sentence and the reader can see by herself which one is the smallest/largest.

line 105: change "sister" to "sister group".

line 108: put the time in mya for the early Jurassic as it is done for Jurassic.

line 212: make the first sentence more comprehensible.

Reviewer 3 Report

Review Report
ijms-2328803-v1
Comparative mitogenomes reveal phylogenic and evolutionary basis of adaptive radiation of lampriform fishes
Jin-fang Wang, Hai-yan Yu, Shao-bo Ma, Qiang Lin, Da-zhi Wang, Xin Wang

230416

Dear authors,
You have determined complete seqs of mt genomes from three rare mesopelagic fish species. You identified unique features in their mt genomes and discussed their evolution from viewpoints of their phylogeny and eco-morphology. It is thus worth publishing your MS in the journal you are submitting to.
Before publication, however, you must address several major issues regarding possible miss-annotation, redundant analysis, back-and-forth arguments in the introduction section, etc.
It contains several typos and grammatical errors. Extensive English editing is definitely necessary.
See below for detail.

L27
adaption radiation -> {adaptation and radiation, adaptive radiation}

L35 introduction
Consider beginning with general statements to draw attention from wider readership. The current delivery is good for lampriform geeks, quite uncommon among readers of the journal.
One point to resolve this issue is to put L60-76 (Mitogenomes have been ..) to the top.

L36
Then, coming "Lampriform species .." (L36-60) to the next needs revision to fit with the above rearrangement beginning with general statements on mtDNA.
The end of the prospecting general statement ".. whole-body endothermy trait" will provoke readers to remind warm-blooded fish.
Then, the next answer may be to put L49-53 to the new 3rd paragraph with reasonable revision to fit in.

L42
the Gadiformes order -> {Gadiformes, the order Gadiformes}

L56
adaptive radiation (grammar) -> adaptively radiated
Grammar will be fine upon revision above, but does "a typical adaptive radiation evolutionary group of fishes" make sense?

L77
three individuals from lampriform fishes (wordy but information deficient) -> three lampriform species

L82
sampling locations for lampriform fishes (imprecise) -> lampriform fish collecting records. Stars indicate sampling localities of this study.

L89
L. incognitus -> Lampris incognitus
Spell-out at the first place independent from the abstract.

L90
R. russelli -> Regalecus russelii
T. ishikawae -> Trachipterus ishikawae

L99
Acanthomorpha was divided into two lineages: Acanthopterygii and Paracanthopterygii
Indicate what is Acanthomorpha in the Figure 3 picture. You may point the acanthomorph basal node with an arrow or bracket the clade entirely.

L107
152.49 million years ago ?
This depend on what "diverged" means. I can see lampriform diverged c.a. 180 MA from other acanthopterygian fish (beryciform through ophidiform, Fig. 3). Do you mean first "divergence" AMONG CROWN GROUPS? But this may overlook possible extinct lampriform STEM or PLESION groups.

L111
outgroup -> an outgroup

L114,132,158
Tables should be supplements. Instead, Fig. S4 should be a main figure.

L117,135,161 tables 1-3 body
Make row height extent.
cox2 might be miss-annotated. It might end with T at 7892 in Lampris, for example.

L119-128
We analyzed the base composition .. showed negative values. -> delete
These tedious long description of your results never work for any of your discussion.

L131
Tables 1, 2 and 3; Supplementary Figure S4 -> Figure X; Supplementary Tables

L136-157
Among all the 13 PCGs, .. encoded on the light (L) strand. (never discussed) -> delete

L161
Intergenic nucleotide data shifted.

L172
RSCU -> Relative synonymous codon usage (RSCU)

L174
Fig. 4B
Replace the left panel with a partial tree from Fig. 3 omitting branch lengths.
Fig. 4C
Two panels are the same.

L176
Relative synonymous codon usage (RSCU) -> RSCU

L178 sub-section 2.3
This is redundant with 2.1. Merge to 2.1 making it shorter.

L194-197
Nucleotide diversity analysis .. morphological characteristics.
Move to introduction or discussion section.

L197
The sliding window analysis -> A sliding window analysis

L209 table 4 body
Vaillantella does have tRD at 7093-7165 in NC_008680 (miss-annotated). Other non-lampriform fishes were also highly probably miss-annotated. Then, this table is eventually the same as Fig. 5B and can safely be omitted.
Make sure with MitoAnnotator (Iwasaki et al 2013). I also suggest you to make sure if some lampriform species really lack tRNAs. While MITOS and ARWEN work on metazoan mt genomes in general and might miss some fish (vertebrate)-specific features, MitoAnnotator is specified for fish.

L220
What are UCEs?

L221
What does "paraphyletic with" mean?
See L105.

L222
Lampriformes diverged approximately 150 million years ago ?
See comment on L107.

L225
Add an appropriate linking word.

L227
Fig. 5B
Replace the left panel with a partial tree from Fig. 3 omitting branch lengths.

L230
above the arrows ? -> in parentheses ?

L266
adaptive changes may have evolved simultaneously with the endothermic traits
This is merely speculation without evidence. Opahs' specific traits are not limited to the endothermy. How about stability or folding efficiency in higher temperatures of the proteins with particular AA substitutions?

L271 Lampriform (English) -> in lower case

L302-304
Phylogenetic analysis .. to construct a phylogenetic tree. -> Phylogenetic analysis was performed based on the 13 PCGs of 67 mitogenomes from 28 teleostean orders including 11 lampriforms with a holostean (Lepisosteus oculatus) grand outgroup.
I think this analysis is enough where non-lampriform fishes work as outgroups. Generally, outgroup rooting is better resolved with a number of outgroups equaling out unwanted evolutionary outliers.

L308
as well as the Shimodaira-Hasegawa-like approximate likelihood-ratio test (never appear in the results) -> delete

L312
Regalecus glesne and Regalecus russelii -> R. russelii and R. glesne
See L90. Put russelii first to save readers' short-term memory spaces on what "R." stands for.

L314
Lepisosteus -> L.
See L98.

L320
relative synonymous codon usage (RSCU) -> RSCU
See L176.

L326
In this study, (verbose) -> delete

L331
the thermostatic traits .. mitochondrial genes (no evidence) -> the opahs with thermostatic traits shared specific amino acid substitutions in mitochondrial protein-coding genes (fact)

L349
Add GenBank accession numbers.

L351 references
Check the reference list carefully again from the beginning. Reference lists are frequently hotbeds of errors. You might add, omit or swap citation in the main text on the way internal revision. Numbering of the references might then shift. If so, readers think you are making irrelevant citation. It is the authors' responsibility that all references are properly cited.

L352
Improper citation. Check thoroughly.

L353,362,etc
Make sure if paper titles are in lower case, Check thoroughly.

L355,etc (many)
Make sure if abbreviated journal title words accompany a dot.

L361
Lampris guttatus -> in Italics
Check thoroughly about Latin names.

L370
Bmc -> BMC
Check thoroughly.

L372
What do square brackets mean?

L374
Sci Rep-UK -> Sci. Rep.

L387
P -> Proc.
Check thoroughly.

L427
Plos -> PLoS
Check thoroughly.

The following item may be helpful for further discussion.

Iwasaki W, Fukunaga T, Isagozawa R, Yamada K, Maeda Y, Satoh TP, Sado T, Mabuchi K, Takeshima H, Miya M, Nishida M. 2013. MitoFish and MitoAnnotator: A mitochondrial genome database of fish with an accurate and automatic annotation pipeline. Mol Biol Evol 30:2531-2540.

Round 2

Reviewer 3 Report

Letter to Authors
ijms-2328803-v2
Phylogenetic and Evolutionary Comparison of Mitogenomes Reveal Adaptive Radiation of Lampriform Fishes
Jin-fang Wang, Hai-yan Yu, Shao-bo Ma, Qiang Lin, Da-zhi Wang, Xin Wang

230502

Dear authors,
I am sorry to see your not well revised v2 MS. I recommended the editor "major" because of (1) back-and-forth nonsensical introduction section, (2) redundant analysis, and (3) issues in comparative mitogenomics.
See below for detail.
Words in braces indicate options. Bracketed words can be omitted. Citing line numbers are of v2 MS. Some lines will shift upon rearrangements.

L34 introduction
You seem unable to see what is "back-and-forth arguments". This section has five paragraphs on (1) mtDNA general, (2) lampriform specifics, (3) mtDNA general, (4) lampriform specifics, (5) what you did. This is a museum specimen of the back-and-forth arguments. Re-order into 1-3-(2+4)-5 with reasonable revision to fit with this rearrangement.

L37-38
The protein coding gene cox1 .. in animals [2]. -> delete
Readers will be embarrassed to see why this inferior matter of short seqs to the whole genome comes next.

L39
Nucleotide diversity analysis -> Nucleotide diversity analysis {along, over} the [mito]genomes
Addition of a pivotal word "genome" emphasizes why mitogenomes in the top sentence.

L42
Nevertheless ?
How does this word work for logical linking?

L50-53
Inverse oarfish and opah.

L50
oarfishes -> oarfishes (Regalecus [spp.])

L51
opahs -> opahs (Lampris [spp.])

L58-60
Most studies .. functional forms. (irrelevant) -> delete

L60-63
The mitogenome is .. phenotypic diversification. -> move to L87

L68-70
Considering .. human mtDNA [16-18]. (irrelevant) -> delete

L71-73
Thus, .. endothermy trait. -> move to L91 or delete

L74-76
Lampriform species .. Regalecidae [19,20]. (verbose) -> delete

L76
Lampriformes systematic analyses are -> Lampriform systematics {with, among} six families (Veliferidae, Lampridae, Lophotidae, Radiicephalidae, Trachipteridae, and Regalecidae) has been

L77
molecular data, and the order -> molecular data. Their classification
Break sentence to make space for L81.

L81
Lampriforms are .. in museums [22]. (back-and-forth) -> move to L77

L35-91
After rearrangement, parse it again making adjustment revision to fit with.

L122
See below.

L125-138
Delete this nonsensical paragraph. Combined with eight previously reported lampriform mitogenomes, you constructed the phylogenetic relationship of lampriforms based on the sequence matrix of 13 PCGs with many teleostean fish and a holostean outgroups. Your analysis described in this paragraph is inferior (data-deficient) to your broader analysis described in Figure 3. Figure S3 should also be omitted.
If you like to talk about node dating among lamoriform families, you may add it at L122 based on your broader analysis. In this case, indicate those families in Figure 3 picture at L103.

L157 Figure 5 picture
Left panel of A
Replace with sub-tree from Figure 3.

L160 table 1 body
Omit Cromeria and Diaphus. These seqs are highly probably incomplete because of difficulty for sequencing. See Lavoue et al 2012 (p 106, L7) and Miya et al 2001 (Table 1 footnote).

L167 Figure 6 picture
Left panel of B
Replace with sub-tree from Figure 3.

L283
R. russelii and Regalecus glesne is 12.2-24.3 million years ?
The node of these two species (R. russelii and R. glesne) is very recent (Figure 3). Is it OK?

L308-314
Make consistent capitalization of family names.

L322 references
This list is still a den of errors. Numbering of the references might shift upon revision. Parse the list carefully again.

L324,etc (many)
Make sure if abbreviated journal title words accompany a dot. Check thoroughly.

L345
B -> Bull.

The following item may be helpful for further discussion.

Lavoue S, Miya M, Moritz T, Nishida M. 2012. A molecular timescale for the evolution of the African freshwater fish family Kneriidae (Teleostei: Gonorynchiformes). Ichthyol Res 59:104-112.

Miya M, Kawaguchi A, Nishida M. 2001. Mitogenomic exploration of higher teleostean phylogenies: a case study for moderate-scale evolutionary genomics with 38 newly determined complete mitochondrial DNA sequences. Mol Biol Evol 18:1993-2009.

Round 3

Reviewer 3 Report

Letter to Authors
ijms-2328803-v3
Phylogenetic and Evolutionary Comparison of Mitogenomes Reveal Adaptive Radiation of Lampriform Fishes
Jin-fang Wang, Hai-yan Yu, Shao-bo Ma, Qiang Lin, Da-zhi Wang, Xin Wang

230510

Dear authors,
This v3 MS seem acceptable after a round of minor revision in the long run. See below for detail.

L4
Wang1,2,3 Hai-yan -> Wang1,2,3, Hai-yan (insert a comma)

L57
spp -> spp. (in Roman)

L59-60
and the oarfishes .. during predation (redundant) -> delete
See the next sentence.

L78-80
for performing .. the phenotypic diversification (verbose) -> phylogeny, classification, and exploring evolutionary footprint related to the phenotypic diversification
Poppulation genetics is out-of-scope here.

L95,100
Make sure a sentence is properly connected.

L102
(insert) This alignment contains eight previously reported lampriform mitogenomes.

L118-121
To better assess .. to date. -> delete
